| 1  | Impact of burial conditions on NO <sub>3</sub> -N source apportionment in groundwater:                                                        |
|----|-----------------------------------------------------------------------------------------------------------------------------------------------|
| 2  | Insights from PCA-APCS-MLR and MixSIAR methods                                                                                                |
| 3  | Yang Liu <sup>1,2</sup> , Jian Luo <sup>3</sup> , Yajie Wu <sup>4</sup> , Ziyang Zhang <sup>1,2</sup> , Xilai Zheng <sup>1,2</sup> , Tianyuan |
| 4  | Zheng <sup>1,2*</sup>                                                                                                                         |
| 5  | 1. College of Environmental Science and Engineering, Ocean University of China,                                                               |
| 6  | Qingdao 266100, China                                                                                                                         |
| 7  | 2. Shandong Provincial Key Laboratory of Marine Engineering Geology and the                                                                   |
| 8  | Environment, Ocean University of China, Qingdao 266100, China                                                                                 |
| 9  | 3. School of Civil and Environmental Engineering, Georgia Institute of Technology,                                                            |
| 10 | Atlanta, GA 30332, USA                                                                                                                        |
| 11 | 4. College of Engineering, Ocean University of China, Qingdao 266100, China                                                                   |
| 12 |                                                                                                                                               |
| 13 | Corresponding author:                                                                                                                         |
| 14 | Tianyuan Zheng                                                                                                                                |
| 15 | E-mail: zhengtianyuan@ouc.edu.cn                                                                                                              |
| 16 |                                                                                                                                               |
| 17 | Abstract                                                                                                                                      |
| 18 | NO3-N contamination in groundwater poses a significant threat to drinking water                                                               |
| 19 | safety and ecosystem health, with accurate source identification being crucial for                                                            |
| 20 | effective pollution control. Previous studies on NO3-N source apportionment in                                                                |
| 21 | groundwater have largely neglected aquifer burial conditions. In this study,                                                                  |
| 22 | groundwater samples from aquifers with varying burial conditions were collected and                                                           |
| 23 | analyzed using an integrated approach combining hydrochemical analysis (PCA-                                                                  |
| 24 | APCS-MLR) and stable isotope mixing modeling (MixSIAR) to identify and quantify                                                               |
| 25 | NO <sub>3</sub> -N pollution sources. The results demonstrate that NO <sub>3</sub> -N concentrations in 75%                                   |
| 26 | of the groundwater samples exceeded the WHO drinking water standard. PCA-APCS-                                                                |
| 27 | MLR analysis revealed that the dominant NO <sub>3</sub> -N sources in unconfined groundwater                                                  |
| 28 | and confined groundwater were chemical fertilizers (52.5%) and manure & sewage                                                                |

(53.9%), respectively. The MixSIAR model further identified soil nitrogen (58%) and 30 manure & sewage (37.9%) as the primary contributors to NO<sub>3</sub>-N in unconfined and 31 confined groundwater, respectively. These findings suggest that unconfined 32 groundwater in regions with high soil nitrogen reserves is at persistent risk of NO<sub>3</sub><sup>-</sup>-N 33 contamination. In addition, neglecting aquifer burial conditions would introduce 34 absolute errors of 22%-24% in source apportionment results obtained from both PCA-35 APCS-MLR and MixSIAR approaches. This study highlights that aquifer confinement 36 must be rigorously considered as a critical factor in NO<sub>3</sub><sup>-</sup>-N source identification and 37 pollution control strategies to enhance the accuracy of source apportionment and the 38 effectiveness of management measures. 39 **Keywords:** Groundwater; NO<sub>3</sub>-N pollution; Source apportionment; PCA-APCS-MLR; 40 MixSIAR

#### 42 **Graphical Abstract**

# NO<sub>3</sub><sup>-</sup>-N source apportionment

Impact of burial conditions

44

# Highlights

- Elucidated the sources of NO<sub>3</sub>-N in aquifers under different burial conditions.
- • Soil nitrogen contributes over 50% to the NO<sub>3</sub>-N in the unconfined aquifer.

- NO<sub>3</sub>-N in confined aquifer mainly originates from manure & sewage.
- Source apportionment results have an error of 24% without considering the burial
- conditions.

#### 1. Introduction

Groundwater NO<sub>3</sub>-N contamination has persisted for nearly a century worldwide, 54 emerging as a critical environmental challenge that threatens both human health and 55 ecological security (Xin et al., 2019). As a highly toxic pollutant, NO<sub>3</sub>-N poses 56 significant health risks including methemoglobinemia and cancer when ingested through drinking water (Picetti et al., 2022), while also causing severe ecological 57 58 impacts such as aquatic eutrophication (Romanelli et al., 2020). The environmental 59 persistence of NO<sub>3</sub>-N is exacerbated by limited natural attenuation in groundwater systems due to weak denitrification processes, resulting in long-term accumulation of 60 61 this contaminant (Rivett et al., 2008). The primary sources of NO<sub>3</sub>-N include non-point 62 source pollution from agricultural activities (fertilizer application and livestock 63 operations) and point source pollution from industrial effluents and domestic sewage 64 (Xin et al., 2021). Consequently, the accurate identification and dissection of NO<sub>3</sub>-N 65 pollution sources are pivotal to the assessment and control of groundwater pollution risks. Despite some advancements in NO<sub>3</sub>-N source apportionment over the past 66 decades (Yang et al., 2013; Gibrilla et al., 2020), the majority of studies have 67 68 overlooked the burial conditions and stratigraphic characteristics of unconfined and 69 confined aquifers. Ignoring this issue can lead to inaccurate source apportionment results, and consequently affect the scientific nature and effectiveness of groundwater 70 71 pollution prevention and control strategies. 72 Current studies on NO<sub>3</sub>-N source apportionment in groundwater predominantly 73 simplifies complex multi-layer aquifer systems into single-layer models without 74 accounting for differences in burial conditions (Yu et al., 2020). While this 75 simplification facilitates analysis, it introduces substantial limitations due to

76 fundamental differences between unconfined and confined aquifers in terms of recharge 77 mechanisms, flow paths, hydraulic characteristics, and contaminant transport behavior. 78 Unconfined aquifers, characterized by strong connectivity with surface water, are 79 highly vulnerable to anthropogenic activities (e.g., agricultural fertilization, industrial 80 effluents, and domestic sewage), allowing contaminants to readily leach into 81 groundwater through precipitation or surface runoff, resulting in rapid NO<sub>3</sub>-N 82 accumulation that typically reflects recent pollution caused by recent human activities 83 (Gutiérrez et al., 2018). In contrast, confined aquifers, protected by overlying aquitards, 84 exhibit slower contaminant migration, with NO<sub>3</sub><sup>-</sup>-N pollution often representing legacy effects from historical agricultural practices (Wong et al., 2015). Failure to differentiate 85 86 these aquifer types may lead to biased source contribution assessments. In addition, the 87 transformation rates of nitrogen components from different pollution sources vary in aquifers with different burial conditions. Unconfined aquifers are generally aerobic 88 89 environments, where the mineralization and nitrification of organic nitrogen occur 90 rapidly, leading to a swift increase in NO<sub>3</sub>-N concentration (Liu et al., 2022). In contrast, 91 confined aquifers tend to have reducing conditions, which restrict the nitrogen 92 transformation rate and cause a lag in NO<sub>3</sub>-N formation (Ma et al., 2019). As a result, 93 the source of NO<sub>3</sub><sup>-</sup>-N may be mistakenly attributed to other pollution sources. Therefore, 94 elucidating the sources of NO<sub>3</sub>-N pollution in actual double-layered aquifers with 95 different burial conditions and revealing the discrepancies between these results and 96 those obtained without considering burial conditions can provide a more accurate basis 97 for groundwater NO<sub>3</sub>-N pollution risk assessment. 98 In recent years, some progress has been made in the identification of NO<sub>3</sub>-N 99 pollution sources in groundwater through the application of hydrochemical analysis 100 methods and stable isotope mixing models (Minet et al., 2017; Yu et al., 2022). 101 Hydrochemical analysis methods mainly include ion ratio methods, hydrochemical 102 diagram methods, and quantitative hydrochemical analysis methods. Among these, 103 quantitative hydrochemical analysis is the core, which encompasses models such as the

https://doi.org/10.5194/egusphere-2025-5482 Preprint. Discussion started: 14 November 2025 © Author(s) 2025. CC BY 4.0 License.

120

123124

chemical mass balance (CMB), positive matrix factorization (PMF), and multivariate statistical models (e.g., principal component analysis and multiple linear regression analysis). Among these methods, the absolute principal component score-multiple linear regression (APCS-MLR) method has garnered considerable attention due to its high efficiency and broad applicability (Meng et al., 2018; Ruan et al., 2024). APCS-MLR can extract key pollution source information by reducing data redundancy through principal component analysis while retaining the essential characteristics of major pollution sources. Additionally, APCS-MLR can establish a quantitative relationship between principal component scores and actual pollutant concentrations via multiple linear regression, thereby accurately calculating the contribution rates of various pollution sources. Subsequently, stable isotope techniques have been applied in the identification of NO<sub>3</sub><sup>-</sup>-N pollution sources in groundwater. The development of this technology in groundwater NO3-N source apportionment has evolved from the use of single isotopes ( $\delta^{15}$ N) to the combined application of multiple isotopes (both  $\delta^{15}$ N and  $\delta^{18}$ O) (Kellman and Hillaire-Marcel, 2003; Ji et al., 2022). By analyzing the isotopic compositions of nitrogen ( $\delta^{15}$ N) and oxygen ( $\delta^{18}$ O) in NO<sub>3</sub><sup>-</sup>-N, this technique can effectively distinguish different sources of NO<sub>3</sub>-N pollution in groundwater (such as agricultural fertilization, domestic sewage, soil nitrogen, and atmospheric deposition) (Ransom et al., 2016), thereby providing an important supplement to traditional hydrochemical analysis methods. To further quantify the contribution proportions of different pollution sources and enhance the accuracy of source identification, the stable isotope mixing model based on the R language, MixSIAR, has been developed. The MixSIAR method, by integrating isotope data with prior information on pollution sources, is capable of quantifying the relative contributions of different pollution sources and assessing the uncertainty of the results. Mao et al. (2023) used the MixSIAR method to analyze the distribution of nitrate pollution sources in the groundwater of Poyang Lake, China, revealing that manure & sewage accounted for 52%, chemical fertilizers for 17%, and soil nitrogen for 21.5% of the pollution sources.

In this study, hydrochemical analysis methods and the MixSIAR method were 133 employed to comprehensively identify the sources of NO<sub>3</sub>-N pollution in aquifers 134 under different burial conditions. 135 The Old County groundwater source area is a vital water supply hub in the central 136 region of Shandong Province. However, with the development of industry and 137 agriculture and the increasing level of urbanization, the Old County source area is 138 facing severe NO<sub>3</sub>-N contamination in groundwater. Identifying the sources of NO<sub>3</sub>-139 N in aquifers under different burial conditions in this region is crucial for elucidating 140 the genesis of "high-nitrogen groundwater". In this study, groundwater samples were 141 collected from 64 wells, and soil, fertilizer, manure, and precipitation samples were also 142 gathered within the study area. The water chemistry indicators and isotopic 143 characteristics of these samples were analyzed. Subsequently, PCA-APCS-MLR and MixSIAR methods were employed for data analysis. The objectives of this study are (1) 144 145 to quantify the concentration and distribution of NO3-N in groundwater within the study area; (2) to quantitatively identify the sources of NO<sub>3</sub>-N contamination in 146 147 aquifers under different burial conditions using hydrochemical analysis and the 148 MixSIAR method; and (3) to define the error in the analysis of groundwater NO<sub>3</sub>-N 149 sources apportionment without considering burial conditions. The study aims to provide 150 a more accurate basis for assessing the risk of NO<sub>3</sub>-N contamination in regional 151 groundwater. 152

# 2. Materials and methods

#### 2.1 Study region

The study area is located on the western edge of the Tai-Lai Basin in the lower reaches of the Yellow River (Fig.1), to the east of Tai'an urban area (117°04′09″E–117°26′45″E, 36°04′16″N–36°12′10″N), with a total area of approximately 220 km². The topography is characterized as a proluvial and alluvial plain at the foot of Mount Tai, with an overall terrain slope from the northwest to the southeast. The study area

163164

176177

falls within the temperate continental semi-humid monsoon climate zone, featuring hot and rainy summers, as well as cold and dry winters. The average annual temperature is 12.9°C, and the average annual precipitation is 790.69 mm. Precipitation exhibits significant spatiotemporal variability, with uneven seasonal distribution and large interannual fluctuations. The primary aquifer formations in the study area consist of two types: the Quaternary unconsolidated porous aquifer group and the Cambrian-Ordovician carbonate rock fracture karst aquifer group. The former is mainly composed of medium to coarse sand, with recharge primarily from atmospheric precipitation and infiltration of surface water, and discharge through evaporation, artificial extraction, replenishment of surface water, and inter-aquifer flow to other aquifers. The latter is mostly situated beneath the Quaternary strata, with recharge mainly from "skylight" recharge of Quaternary water and lateral flow recharge from regional bedrock fracture aquifers, and discharge through artificial extraction, runoff discharge, and upward replenishment to the Quaternary porous water. The urban population in the study area is approximately 28,000, with over 85% of the population engaged in agriculture and animal husbandry.

**Fig.1.** Location of the Tailai Basin in lower reaches of the Yellow River and sampling sites in the study region.

189

195

205206

## 2.2 Sample collection

A total of 64 groundwater samples were collected from the study area. Prior to sampling, wells were thoroughly flushed, and samples were taken from a depth of more than 0.5 m below the groundwater table. For sealed wells, water stored in the pumping pipe was completely drained before sampling. After collection, groundwater samples were filtered through a 0.45 µm membrane filter and stored in 500 mL amber glass bottles, which were then sealed and transported to the laboratory for refrigeration at 4°C. Groundwater samples intended for isotopic analysis were filtered through a 0.22 μm membrane filter and stored frozen in 50 mL polyethylene bottles. Five atmospheric precipitation samples were collected using stainless-steel precipitation samplers. For single-day precipitation events, one complete-event sample was collected, while for multi-day precipitation events, samples were collected at 24-hour intervals. All precipitation samples were stored in polyethylene bottles. Five typical fertilizer samples (including urea and compound fertilizers) were collected based on local farmers' fertilization practices. Given the difficulty in distinguishing between manure & sewage pollution sources using  $\delta^{15}N$  and  $\delta^{18}O$  isotopes, these two sources were combined into one category in this study. A total of 10 samples (including cow manure, pig manure, chicken manure, sheep manure, goose manure, and sewage) were collected. Manure samples were air-dried for later use, while sewage samples were filtered through a 0.22 μm membrane filter and stored frozen. Additionally, 20 agricultural soil samples were collected using the plum blossom point layout method. Each sample was composed of a mixture from 5 to 15 sampling points at a depth of 30 cm, with all sampling points avoiding fertilized areas. The collected soil samples were thoroughly mixed after removing roots and gravel and then stored.

#### 2.3 Sample Analysis

The concentration of NO<sub>3</sub><sup>-</sup>-N was determined using ultraviolet spectrophotometry.

The concentrations of K<sup>+</sup>, Na<sup>+</sup>, Ca<sup>2+</sup>, Mg<sup>2+</sup>, Cl<sup>-</sup>, and SO<sub>4</sub><sup>2-</sup> were measured using an ion chromatograph (ICS-3000, Dionex, USA), the concentration of HCO<sub>3</sub><sup>-</sup> was determined

by acid-base titration. For liquid samples (groundwater, atmospheric precipitation, and sewage),  $\delta^{15}N$  and 208 δ<sup>18</sup>O were measured using the azide reduction method. This involved chemically 209 reducing NO<sub>3</sub>-N in the samples to N<sub>2</sub>O, which was then analyzed using an elemental 210 211 analyzer coupled with an isotope ratio mass spectrometer (Vario Isotope Cube -Isoprime, Elementar) to obtain the isotopic values of  $\delta^{15}$ N and  $\delta^{18}$ O. For solid samples 212 (soil, fertilizer, and manure),  $\delta^{15}$ N and  $\delta^{18}$ O were measured using the high-temperature 213 oxidation method. This procedure involved weighing an appropriate amount of 214 215 thoroughly ground powder sample, encapsulating it in a tin cup, and analyzing it using 216 an elemental analyzer coupled with an isotope ratio mass spectrometer. 217 2.4 Source apportionment methods 218 2.4.1 Hydrochemical analysis method 219 (1) Piper diagram 220 The method used to determine the hydrochemical type of groundwater is the Schoeller classification method. First, the concentrations of K<sup>+</sup>, Na<sup>+</sup>, Ca<sup>2+</sup>, Mg<sup>2+</sup>, HCO<sub>3</sub><sup>-</sup>, 221 SO<sub>4</sub><sup>2</sup>-, Cl<sup>-</sup>, and NO<sub>3</sub><sup>-</sup>-N in groundwater samples, expressed in milligrams per liter (mg 222 L<sup>-1</sup>), are converted to milliequivalent concentrations (meg L<sup>-1</sup>). Subsequently, the 223 224 milliequivalent percentage of each ion is calculated. Finally, the hydrochemical type is 225 determined based on the ions with a milliequivalent percentage greater than 25%. The 226 milliequivalent percentages of cations and anions for all water samples in the water 227 quality monitoring data are plotted on a Piper diagram. 228 (2) PCA-APCS-MLR 229 Principal Component Analysis (PCA) was employed to extract the dominant pollution factors, and the potential sources of groundwater contamination were inferred 230 231 in conjunction with water quality indicators:

$$\begin{cases}
PC_{1}=\mu_{11}x_{1}+\mu_{12}x_{2}+\cdots+\mu_{1j}x_{j} \\
PC_{2}=\mu_{21}x_{1}+\mu_{22}x_{2}+\cdots+\mu_{2j}x_{j} \\
\vdots \\
PC_{m}=\mu_{m1}x_{1}+\mu_{m2}x_{2}+\cdots+\mu_{mr}x_{j}
\end{cases} (1)$$

- PC<sub>1</sub>, PC<sub>2</sub>, ..., PC<sub>m</sub> represent the principal components 1, 2, ..., m that can explain the
- original indicators. The eigenvalues  $\lambda_m$  ( $m \le j$ ) of the correlation coefficient matrix are
- the variances of  $PC_m$ , and the larger the variance, the greater the contribution to the
- principal component.
- Subsequently, on the basis of PCA, the absolute principal component scores (APCS)
- were determined. A multiple linear regression (MLR) was performed with the measured
- pollutant concentrations as the dependent variables and the absolute principal
- component scores as the independent variables. The pollution contributions of each
- factor were calculated based on the regression coefficients, thereby determining the
- contribution rates of the pollution sources:

(A<sub>0</sub>)<sub>p</sub> = 
$$\sum_{j=1}^{J} S_{pj} (Z_0)_j$$
 (2)

- p represents the principal component extracted during the principal component analysis
- (PCA) process.  $(A_0)_p$  denotes the absolute principal component score for principal
- component p.  $S_{pj}$  represents the scoring coefficient of indicator j within principal
- component p.

$$C_{j} = b_{j} + \sum_{p=1}^{P} b_{pj} \times APCS_{ip}$$
 (3)

- $C_j$  represents the measured concentration of pollutant j.  $b_j$  denotes the constant term in
- the multiple linear regression analysis.  $b_{pj}$  represents the regression coefficient for
- principal component p.  $b_{pj} \times APCS_{ip}$  indicates the concentration contribution of principal
- component p to pollutant j in sample i. The average value of  $b_{pj} \times APCS_{ip}$  represents the
- average concentration contribution of principal component p (the pollution source) to

- pollutant j. Finally, by converting the concentration contributions of each pollution
- source into percentages, the contribution rates of the pollution sources can be
- determined.
- 2.4.2 MixSIAR method
- The principle of the MixSIAR method is to use the Dirichlet distribution as the prior
- distribution and to obtain the posterior distribution characteristics of the contributions,
- such as the mean, variance, and probability density, through the application of Bayes'
- theorem. Assuming there are n samples, k different sources, and j isotopes, the
- MixSIAR mixing model can be expressed as follows:

$$X_{ij} = \sum_{k=1}^{K} P_k (S_{jk} + \varepsilon_{jk}) + V_{ij}$$

$$S_{ik} \sim N(\mu_{ik}, \omega_{ik}^2)$$

$$\varepsilon_{ik} \sim N(\lambda_{ik}, \tau_{ik}^2)$$

$$v_{ii} \sim N(0, \sigma_i^2) \tag{4}$$

- $X_{ij}$  represents the value of the j isotope in the i sample (i=1, 2, 3, ..., N; j=1, 2, 3, ...,
- 268 J).  $P_k$  denotes the contribution rate of the k source (k=1, 2, 3, ..., K), which is predicted
- using the MixSIAR method.  $S_{jk}$  represents the value of the j isotope from the k source,
- with a mean of  $\mu_{jk}$  and a variance of  $\omega_{jk}^2$ .  $\varepsilon_{jk}$  represents the enrichment coefficient of the
- j isotope from the k source, with a mean of  $\lambda_{jk}$  and a variance of  $\tau_{jk}^2$ .  $\nu_{ij}$  represents the
- residual, with a mean of 0 and a variance of  $\sigma_j^2$ .

#### 2.5 Data analysis

- The stable isotope mixing model used in this study was run in the R package
- MixSIAR (R version x64 4.3.2). The Pearson correlation test was employed to evaluate
- the relationships between hydrochemical indices, with data analysis conducted using
- SPSS 20. The spatial distribution of NO<sub>3</sub>-N concentrations was generated using Surfer
- 15 software, and the cartographic work was completed with Origin 2020.

#### 3. Results

#### 3.1 Characteristics of groundwater NO<sub>3</sub>-N pollution

The type of groundwater in the study area is predominantly of the Ca-type, with the molar percentage of Ca<sup>2+</sup> exceeding 50% in most sampling points (Fig.2). In addition, the groundwater in the study area can be classified into two main types: Cl<sup>-</sup>·NO<sub>3</sub><sup>-</sup>·HCO<sub>3</sub><sup>-</sup>-Ca<sup>2+</sup> and Cl<sup>-</sup>·NO<sub>3</sub><sup>-</sup>·SO<sub>4</sub><sup>-</sup>-Ca<sup>2+</sup>. Specifically, the Cl<sup>-</sup>·NO<sub>3</sub><sup>-</sup>·HCO<sub>3</sub><sup>-</sup>-Ca<sup>2+</sup> type is primarily found in karst water, while the Cl<sup>-</sup>·NO<sub>3</sub><sup>-</sup>·SO<sub>4</sub><sup>-</sup>-Ca<sup>2+</sup> type is mainly distributed in pore water.

 $\textbf{Fig.2.} \ \textbf{Piper} \ \textbf{graph} \ \textbf{illustrating} \ \textbf{hydrochemical} \ \textbf{types} \ \textbf{of} \ \textbf{groundwater}.$ 

Kriging interpolation was employed to analyze the spatial distribution of  $NO_3$ <sup>-</sup>-N concentration in the groundwater of the study area. The results indicate that the  $NO_3$ <sup>-</sup>-N concentration in the groundwater ranges from 0 to 68 mg N L<sup>-1</sup>, with an average concentration of 22.45 mg N L<sup>-1</sup> (Fig.3). Based on the World Health Organization's drinking water standard ( $NO_3$ <sup>-</sup>-N  $\leq 10$  mg N L<sup>-1</sup>), the  $NO_3$ <sup>-</sup>-N exceedance rate in the study area is 75%, indicating a relatively severe overall pollution status. Specifically, the  $NO_3$ <sup>-</sup>-N concentration in unconfined groundwater ranges from 0 to 68 mg N L<sup>-1</sup>, with an average concentration of 29.9 mg N L<sup>-1</sup>, while that in confined groundwater ranges from 0 to 62.1 mg N L<sup>-1</sup>, with an average concentration of 20.1 mg N L<sup>-1</sup>. Additionally, 50% of the sampling sites in unconfined groundwater and 14% in confined groundwater exceed 30 mg N L<sup>-1</sup> (Class V groundwater quality standard in China), suggesting that  $NO_3$ <sup>-</sup>-N pollution in unconfined groundwater is more severe

306307

312313

than that in confined groundwater. Spatially, the NO<sub>3</sub>-N pollution in the groundwater exhibits significant spatial heterogeneity, with the central part of the study area experiencing more severe NO<sub>3</sub>-N contamination compared to the western and eastern regions.

**Fig.3.** (a) Spatial distribution map of NO<sub>3</sub><sup>-</sup>-N concentrations in unconfined and confined groundwater of the study region. (b) Boxplot of NO<sub>3</sub><sup>-</sup>-N concentrations. The dot and line represent mean value and median. (c) Percentages of NO<sub>3</sub><sup>-</sup>-N concentrations in unconfined groundwater and confined groundwater (<10 mg N L<sup>-1</sup>, ranging from 10 to 30 mg N L<sup>-1</sup>, and >50 mg N L<sup>-1</sup>).

# 3.2 NO<sub>3</sub>-N sources apportionment by PCA-APCS-MLR model

### 3.2.1 Qualitative identification of NO<sub>3</sub>-N sources

The results of Pearson correlation analysis demonstrate that, in the generalized single-aquifer layer without consideration of aquifer burial conditions (hereinafter

referred to as the generalized single-aquifer layer) (Fig.4a), there is a strong correlation among the nine hydrochemical indicators. For example, Mg<sup>2+</sup> is strongly correlated with Na<sup>+</sup>, Ca<sup>2+</sup>, Cl<sup>-</sup>, SO<sub>4</sub><sup>2-</sup>, HCO<sub>3</sub><sup>-</sup>, and NO<sub>3</sub><sup>-</sup>, while NO<sub>3</sub><sup>-</sup> exhibits strong correlations with Ca<sup>2+</sup>, Mg<sup>2+</sup>, and Cl<sup>-</sup>. In the actual double-aquifer layer where aquifer burial conditions are taken into account (hereinafter referred to as the actual double-aquifer layer) (Fig.4b and Fig.4c), the indicators also show strong correlations. Specifically, Ca<sup>2+</sup> is strongly correlated with Na<sup>+</sup>, Mg<sup>2+</sup>, Cl<sup>-</sup>, SO<sub>4</sub><sup>2-</sup>, HCO<sub>3</sub><sup>-</sup>, and NO<sub>3</sub><sup>-</sup>, and NO<sub>3</sub><sup>-</sup> displays strong correlations with DO, Ca<sup>2+</sup>, Mg<sup>2+</sup>, and Cl<sup>-</sup>. Therefore, the selected hydrochemical indicators are suitable for principal component analysis.

**Fig.4.** Pearson correlation analysis of different hydrochemical indexes. (a) Generalized single-layer aquifer. (b) Actual double-layer aquifer (unconfined groundwater). (c) Actual double-layer aquifer (confined groundwater).

Subsequently, we calculated the rotated factor loadings using the varimax rotation

method. The factor loadings reflect the relative importance of each variable in the principal components. Typically, factor loadings greater than 0.7, between 0.7 and 0.5, and between 0.5 and 0.3 are defined as strong, moderate, and weak loadings, respectively. Based on these factor loading results, we identified pollution sources. The results indicate that, for the generalized single-aquifer layer (Fig.5a), P1 represents pollution from chemical fertilizers, P2 represents natural sources, and P3 represents pollution from manure & sewage. For the actual double-aquifer layer, in the unconfined groundwater, P1 represents natural sources, P2 represents pollution from chemical fertilizers, and P3 represents pollution from manure & sewage. In the confined groundwater, P1 represents pollution from chemical fertilizers, P2 represents pollution from manure and domestic sewage, and P3 represents natural sources.

**Fig.5.** Sankey graph of rotation factor load matrix for hydrochemical indexes. (a) Generalized single-layer aquifer. (b) Actual double-layer aquifer (unconfined groundwater). (c) Actual double-

layer aquifer (confined groundwater).

#### 3.2.2 Quantitative apportionment of NO<sub>3</sub>-N sources

Following the qualitative identification of the major pollution sources, the APCS-MLR method was employed to quantitatively analyze the pollution sources (Table 1). For the generalized single-aquifer layer, the regression equation between NO<sub>3</sub>-N concentration and the absolute principal component scores was established as:  $C=7.231\times P1-9.786\times P2+5.655\times P3-4.45$  ( $R^2=0.789$ , p<0.01). This regression model explains 78.9% of the variation in NO<sub>3</sub>-N concentration, with the remaining 21.1% attributable to unknown pollution sources. For the actual double-aquifer layer, in the unconfined aquifer, the regression equation between NO<sub>3</sub>-N concentration and the absolute principal component scores is:  $C=6.85\times P1+17.84\times P2+3.78\times P3+3.197$  ( $R^2=0.838$ , p<0.01), explaining 83.8% of the variation in NO<sub>3</sub>-N concentration, and the remaining 16.2% is attributed to unknown pollution sources. In the confined aquifer, the regression equation is:  $C=5.12\times P1+9.16\times P2-1.74\times P3-9.26$  ( $R^2=0.841$ , p<0.01), accounting for 84.1% of the variation in NO<sub>3</sub>-N concentration, with the remaining 15.9% attributed to unknown pollution sources.

 Table 1. Multiple regression equation based on APCS-MLR.

| Aquifers                                      | Multiple regression equation      |
|-----------------------------------------------|-----------------------------------|
| Single-layer aquifer                          | C=7.231×P1-9.786×P2+5.655×P3-4.45 |
| Double-layer aquifer (unconfined groundwater) | C=6.85×P1+17.84×P2+3.78×P3+3.197  |
| Double-layer aquifer (confined groundwater)   | C=5.12×P1+9.16×P2-1.74×P3-9.26    |

Furthermore, we calculated the contribution rates of each pollution source using the regression equations (Fig.6). For the generalized single-aquifer layer, the contribution rates of chemical fertilizers, manure & sewage, natural sources, and unknown pollution sources were 48.75%, 30.15%, 0%, and 21.1%, respectively, with chemical fertilizers being the dominant pollution source. For the actual double-aquifer layer, in the unconfined groundwater, the contribution rates of chemical fertilizers, manure & sewage, natural sources, and unknown pollution sources were 52.51%, 11.13%, 20.16%,

and 16.2%, respectively. In the confined groundwater, the contribution rates were 30.15% for chemical fertilizers, 53.95% for manure & sewage, 0% for natural sources, and 15.9% for unknown pollution sources. Chemical fertilizers and manure & sewage were identified as the primary pollution sources in the unconfined and confined groundwater, respectively.

Fig.6. Quantitative apportionment of NO<sub>3</sub>-N source based on the PCA-APCS-MLR method

#### 3.3 NO<sub>3</sub>-N sources apportionment by MixSIAR model

# 3.3.1 Distribution characteristics of $\delta^{15}N$ and $\delta^{18}O$ in groundwater

We analyzed the  $\delta^{15}N$  and  $\delta^{18}O$  values of  $NO_3$ -N in potential pollution sources (atmospheric deposition, soil nitrogen, chemical fertilizers, and manure & sewage) as well as in groundwater within the study area. The results of the  $\delta^{15}N$  and  $\delta^{18}O$  values for the potential pollution sources are presented in the Supplementary data (S1). The  $\delta^{15}N$  and  $\delta^{18}O$  values of  $NO_3$ -N in groundwater within the study area are shown in Fig.7. For the generalized single-aquifer layer, the  $\delta^{15}N$  values range from 2.8‰ to 29.29‰, with an average of 9.85‰, while the  $\delta^{18}O$  values range from -0.85‰ to 15.12‰, with an average of 4.42‰. For the actual double-aquifer layer, the average  $\delta^{15}N$  and  $\delta^{18}O$  values in unconfined groundwater are 10.16‰ and 3.93‰, respectively, and in confined groundwater, the average  $\delta^{15}N$  and  $\delta^{18}O$  values are 9.71‰ and 4.6‰, respectively.

Fig.7. Spatial distribution of  $\delta^{15}$ N-NO<sub>3</sub> (a) and  $\delta^{18}$ O-NO<sub>3</sub> (b) in the groundwater

# 3.3.2 Qualitative identification of NO<sub>3</sub>-N sources

The NO<sub>3</sub><sup>-</sup>-N in the groundwater of the study area originates from multiple nitrogen pollution sources. Given the distinct isotopic signatures of  $\delta^{15}$ N and  $\delta^{18}$ O of NO<sub>3</sub><sup>-</sup>-N from different sources, qualitative identification of groundwater NO<sub>3</sub><sup>-</sup>-N sources can be achieved based on the characteristic ranges of these dual isotopes. As shown in Fig.8, the majority of the  $\delta^{15}$ N and  $\delta^{18}$ O values in groundwater locate within the ranges characteristic of chemical fertilizers, soil nitrogen, and manure & sewage. This indicates that the NO<sub>3</sub><sup>-</sup>-N in the groundwater of the study area is primarily derived from these three pollution sources.

**Fig.8.** Isotopic ratio plot of  $\delta^{15}$ N and  $\delta^{18}$ O of NO<sub>3</sub>-N in Groundwater

## 3.3.3 Quantitative apportionment of NO<sub>3</sub>-N sources

The  $\delta^{15}N$  and  $\delta^{18}O$  values of groundwater samples, as well as the mean values and standard deviations of  $\delta^{15}N$  and  $\delta^{18}O$  for potential pollution sources, were used as known parameters and input into the MixSIAR method. To account for potential errors caused by isotopic fractionation, we calculated the fractionation coefficients for  $\delta^{15}N$  and  $\delta^{18}O$  of different pollution sources (Supplementary data, S2) and incorporated these coefficients into the MixSIAR method. Ultimately, by treating the contribution rates of different pollution sources as random variables, we established probabilistic distribution equations for pollution source contributions using the MixSIAR method, thereby determining the extent to which each pollution source contributes to  $NO_3$ -N pollution in groundwater. The results indicate that, for the generalized single-aquifer layer (Fig.9a), the contribution rates of atmospheric deposition, soil nitrogen, chemical fertilizers, and manure & sewage to  $NO_3$ -N pollution are 4.6%, 49.5%, 27.8%, and 18.1%, respectively. For the actual double-aquifer layer (Fig.9b), in the unconfined groundwater, the contribution rates of atmospheric deposition, soil nitrogen, chemical fertilizers, and manure & sewage to  $NO_3$ -N pollution are 5.7%, 58%, 20.1%, and

- 16.2%, respectively. In the confined groundwater, the contribution rates of these four
- pollution sources are 3.1%, 27.5%, 31.5%, and 37.9%, respectively.

Fig.9. Quantitative apportionment of NO<sub>3</sub>-N source based on the MixSIAR method. (a)

Generalized single-layer aquifer. (b) Actual double-layer aquifer.

**4. Discussion** 

We employed both the PCA-APCS-MLR method and the MixSIAR method to quantitatively identify the sources of NO<sub>3</sub>-N in aquifers under different burial conditions. For the PCA-APCS-MLR analysis, different ions exhibit varying loading strengths in each principal component. Therefore, through hydrochemical analysis and statistical methods, we can calculate and infer the type of pollution source represented by each principal component. For example, in unconfined groundwater, Na<sup>+</sup>, Ca<sup>2+</sup>, Mg<sup>2+</sup>, HCO<sub>3</sub>-, SO<sub>4</sub><sup>2-</sup>, and Cl<sup>-</sup> have strong loadings in P1. These ions are all major ions in groundwater, and their average concentrations are relatively low. Moreover, correlation analysis results show that the concentration of NO<sub>3</sub>-N has very low correlation with the concentrations of Na<sup>+</sup>, Mg<sup>2+</sup>, HCO<sub>3</sub>-, SO<sub>4</sub><sup>2-</sup>, and Cl<sup>-</sup>, indicating that NO<sub>3</sub>-N does not originate from the same source as these ions (Yu et al., 2022). Thus, it is demonstrated that P1 represents a natural source. In P2, Ca<sup>2+</sup> and NO<sub>3</sub>-N have

439440

446447

457458

strong loadings. The correlation results (Fig.4) indicate a significant positive correlation (p < 0.01) between Ca<sup>2+</sup> and NO<sub>3</sub>-N, suggesting that Ca<sup>2+</sup> originates from anthropogenic pollution. This is because calcium is required in the cultivation of tomatoes and cucumbers (the main crop types in the study area) (Gulbagca et al., 2020), and the extensive use of calcium fertilizers during the application of base fertilizers and top-dressing fertilizers also increases the concentration of Ca<sup>2+</sup> in groundwater (Schot and Wassen, 1993). Therefore, P2 primarily represents the pollution source from chemical fertilizers. In P3, DO has a strong loading. Since the oxidation and decomposition of organic matter require a large amount of DO (Díaz-Cruz and Barceló, 2008), the strong loading of DO is associated with organic pollution of groundwater (such as from manure and domestic sewage). Thus, P3 mainly represents the pollution sources of manure & sewage. After determining the pollution sources represented by each principal component using the above methods, we can calculate the contribution rate of each pollution source using regression equations. The PCA-APCS-MLR method has the advantages of being rapid and convenient, but it has the disadvantage of being unable to further identify soil nitrogen as a pollution source. To compensate for this limitation, the MixSIAR method was further employed to analyze the sources of pollution. We identified soil nitrogen as another important source of NO<sub>3</sub>-N in groundwater. Additionally, we incorporated isotope fractionation coefficients into the calculations. This is because NO<sub>3</sub>-N from different sources (atmospheric deposition, soil nitrogen, chemical fertilizers, and manure & sewage) has distinct isotopic signatures, and isotopic fractionation occurs during the transport and transformation processes of nitrogen in the groundwater system (such as ammonification and nitrification), leading to changes in the  $\delta^{15}N$  and  $\delta^{18}O$  values of  $NO_3$ -N (Shu et al., 2024). Therefore, considering the effect of isotope fractionation can better eliminate uncertainties in nitrogen transformation processes and significantly improve the accuracy of source apportionment results. This approach has also been confirmed by previous studies (Yu et al., 2020).

473474

476477

In this study, the PCA-APCS-MLR method identified chemical fertilizers as the primary source of NO<sub>3</sub>-N in unconfined groundwater and manure & sewage as the main sources of NO<sub>3</sub>-N in confined groundwater. The MixSIAR method further revealed that soil nitrogen is a dominant pollution source for unconfined groundwater, with a higher contribution rate than that of chemical fertilizers. For confined groundwater, MixSIAR also confirmed that manure & sewage are the major sources of NO<sub>3</sub>-N. The findings for unconfined groundwater can be attributed to the extensive use of chemical fertilizers in agricultural production (Hao et al., 2025). Nitrogen from these fertilizers can directly leach into the unconfined aquifer, causing NO<sub>3</sub>-N pollution. Additionally, excess nitrogen accumulates in the soil and vadose zone, where it is transformed from organic nitrogen to NH<sub>4</sub><sup>+</sup>-N and then to NO<sub>3</sub><sup>-</sup>-N under the action of soil microorganisms (Liu et al., 2023). While NH<sub>4</sub><sup>+</sup>-N can be adsorbed and immobilized by the soil, NO<sub>3</sub>-N can leach into the deeper vadose zone and aquifer through atmospheric precipitation or agricultural irrigation, directly contaminating unconfined groundwater (Wan et al., 2024). Therefore, in assessing the sources of NO<sub>3</sub> -N pollution in regional groundwater, it is crucial not only to focus on the application rates of chemical fertilizers but also to pay attention to the storage of nitrogen in the soil and vadose zone. These accumulated nitrogen compounds can continuously leach into unconfined groundwater under external disturbances (such as irrigation or precipitation), leading to persistent contamination (Niu et al., 2022). Therefore, it is essential to guide local farmers in implementing surface management practices (such as the use of chemical fertilizers and the application of manure) to enforce optimal agricultural irrigation policies, including reducing irrigation frequency, to delay the transport of stored nitrogen in the soil to the aguifer. Regarding the results for confined groundwater, the nitrogen in manure/ sewage primarily exists in the form of large molecules. These complex nitrogen compounds are difficult to degrade microbially or transform chemically in a short period, leading to their long-term persistence in the environment. These pollutants can enter surface water bodies through surface runoff or

490 infiltration and then gradually transport to deeper aquifers via the interflow recharge 491 process between unconfined and confined aquifers, resulting in persistent 492 contamination (McDonough et al., 2022). Therefore, for the prevention and control of NO<sub>3</sub>-N pollution in confined aquifers, it is crucial to focus on the source control of 493 494 manure & sewage to block the migration pathways of pollutants and mitigate their long-495 term impacts on confined aquifers. 496 This study compared the errors in source apportionment of NO<sub>3</sub>-N in aquifers with 497 and without consideration of burial conditions. The absolute errors for the PCA-APCS-498 MLR method were 4%–20% and 5%–24%, while those for the MixSIAR method were 499 1.1%-8.5% and 1.5%-22%. The causes of these errors can be attributed to two main 500 factors. First, the sources and recharge mechanisms of groundwater in unconfined and 501 confined aquifers differ significantly, leading to distinct isotopic compositions and characteristic values. For example, the isotopic signature of a pollution source in an 502 503 unconfined aquifer may resemble that of another source in a confined aquifer. When 504 mixed calculations are performed without considering the actual burial conditions, the 505 isotopic differences are obscured, resulting in confusion in pollution source 506 identification, inaccurate contribution rate calculations, and incomplete analysis of 507 pollution processes. This, in turn, may lead to underestimation or overestimation of the 508 contributions of pollution sources to groundwater under different burial conditions. 509 Second, the migration and transformation capacities of nitrogen vary among different 510 geological strata. Hydrogeological conditions can influence the intensity of 511 biogeochemical processes such as ammonification, nitrification, denitrification, and adsorption (Huang et al., 2022; Li et al., 2023), which further alter NO<sub>3</sub><sup>-</sup>-N 512 concentrations and isotopic signatures. This ultimately affects the accuracy and 513 514 reliability of pollution source apportionment. Consequently, pollution control measures 515 may deviate from actual needs and fail to effectively mitigate and reduce NO3-N 516 contamination in groundwater.

#### 5. Conclusion

525526

The study investigated the sources of NO<sub>3</sub>-N pollution in aquifers under different burial conditions and analyzed the errors in source apportionment results of NO<sub>3</sub>-N pollution in groundwater when burial conditions were not considered. The results showed that the groundwater NO<sub>3</sub>-N concentration in the study area ranged from 0 to 68 mg N L<sup>-1</sup>, with an exceedance rate of 75%. The NO<sub>3</sub>-N pollution in unconfined groundwater (average concentration 29.9 mg N L-1) was more severe than that in confined groundwater (average concentration 20.1 mg N L<sup>-1</sup>). The PCA-APCS-MLR method confirmed that the chemical fertilizer is the primary source of NO<sub>3</sub>-N in unconfined groundwater, while the MixSIAR method further identified soil nitrogen as the main source of NO<sub>3</sub>-N pollution in unconfined groundwater, with a higher contribution rate than that of chemical fertilizers. Therefore, it is necessary to focus on the storage of nitrogen in the soil and improve agricultural irrigation practices to prevent rapid infiltration of NO<sub>3</sub>-N into unconfined groundwater, which could lead to persistent contamination. Both analytical methods indicated that manure & sewage are the main sources of NO<sub>3</sub>-N in confined groundwater. When the burial conditions of groundwater were not considered, both methods yielded significant errors (with absolute errors reaching up to 24%). Thus, to accurately identify and effectively manage the sources of NO<sub>3</sub>-N pollution in groundwater, it is essential to carefully incorporate the actual burial conditions of regional aquifers into the analysis.

537538

#### Acknowledgments

- This work was supported by the National Natural Science Foundation of China (No.
- 42422207).

542543

# CRediT authorship contribution statement

- Y L: Writing review & editing, Writing original draft, Visualization, Methodology,
- Investigation, Formal analysis, Data curation, Conceptualization.

- **J** L: Writing review & editing, Supervision, Methodology, Conceptualization.
- YJ W: Writing review & editing, Supervision, Methodology, Conceptualization.
- **ZY Z**: Visualization, Investigation, Methodology, Conceptualization.
- **XL Z**: Supervision, Conceptualization.
- TY Z: Writing review & editing, Supervision, Resources, Methodology, Investigation,
- Conceptualization, Funding acquisition.

# 553 Declaration of competing interest

- The authors declare that they have no known competing financial interests or
- personal relationships that could have appeared to influence the work reported in this
- paper.

# 557

552

#### 558 Data availability statement

- The data of this study can be found in Liu (2025), "Data Availability for Water
- Resources Research", Mendeley Data, V1, doi: 10.17632/53d3ktbg8d.1.

# 561

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

| 630 | nitrification 15N-enrichment factor in a drinking water source region. Sci. Total Environ. 918,           |
|-----|-----------------------------------------------------------------------------------------------------------|
| 631 | 170617.                                                                                                   |
| 632 | Wan, Y., Li, R., Yao, K., Peng, C., Wang, W., Li, N., Wang, X., 2024. Bioelectro-barriers prevent nitrate |
| 633 | leaching into groundwater via nitrogen retention. Water Res. 249, 120988.                                 |
| 634 | Wong, W.W., Grace, M.R., Cartwright, I., Cook, P.L., 2015. Unravelling the origin and fate of nitrate in  |
| 635 | an agricultural-urban coastal aquifer. Biogeochemistry 122, 343-360.                                      |
| 636 | Xin, J., Liu, Y., Chen, F., Duan, Y., Wei, G., Zheng, X., Li, M., 2019. The missing nitrogen pieces: A    |
| 637 | critical review on the distribution, transformation, and budget of nitrogen in the vadose zone-           |
| 638 | groundwater system. Water Res. 165, 114977.                                                               |
| 639 | Xin, J., Wang, Y., Shen, Z., Liu, Y., Wang, H., Zheng, X., 2021. Critical review of measures and decision |
| 640 | support tools for groundwater nitrate management: A surface-to-groundwater profile perspective. J.        |
| 641 | Hydrol. 598, 126386.                                                                                      |
| 642 | Yang, L., Han, J., Xue, J., Zeng, L., Shi, J., Wu, L., Jiang, Y., 2013. Nitrate source apportionment in a |
| 643 | subtropical watershed using Bayesian model. Sci. Total Environ. 463, 340-347.                             |
| 644 | Yu, L., Zheng, T., Yuan, R., Zheng, X., 2022. APCS-MLR model: a convenient and fast method for            |
| 645 | quantitative identification of nitrate pollution sources in groundwater. J. Environ. Manage. 314,         |
| 646 | 115101.                                                                                                   |
| 647 | Yu, L., Zheng, T., Zheng, X., Hao, Y., Yuan, R., 2020. Nitrate source apportionment in groundwater        |
| 648 | using Bayesian isotope mixing model based on nitrogen isotope fractionation. Sci. Total Environ.          |
| 649 | 718, 137242.                                                                                              |
| 650 |                                                                                                           |