# Peer review of "(untitled)"

_EGUsphere, 2025_

## Referee Comment (RC1)

General comments:

Contamination of groundwater systems by high nitrogen emissions from agricultural land use are one of the most common and serious problems of water resources management and research at the global scale. Thus the paper addresses a highly relevant problem. The manuscript combines hydrochemical analysis (PCA-APCS-MLR) and stable isotope mixing modeling (MixSIAR) to identify and quantify $NO_3^-$-N pollution sources in groundwater for unconfined and confined aquifers. It compares the result of computing the apportionment, assuming the two aquifers as a single water body, or as two water bodies. The argument for such comparison is that previous studies on NO3- -N source apportionment have largely neglected aquifer type. I tend to accept it after appropriate revisions.

Detailed comments:

1. The term "burial conditions" is not appropriate in this context. Please consider using more precise hydrogeological terminology, such as "groundwater occurrence conditions", to better reflect the intended meaning.

2. The manuscript uses several abbreviations without introducing their full forms upon first use. The full form of each abbreviation should be provided at its first occurrence (e.g., Nitrate-N ($NO_3^-$-N)).

3. Sample-testing and correlation methods lack procedural detail and citations. Please add detailed information.

4. L. 235: Add citation to the original work by Thurston and Spengler (1985) (https://doi.org/10.1016/0004-6981(85)90132-5) who proposed the APCS-MLR method.

5. L. 256: Add citation to Moore and Semmens (2008) (DOI: 10.1111/j.1461-0248.2008.01163.x) and Stock and Semmens (2016)

(doi:10.5281/zenodo.1209993.)for the MixSIAR method and R code.

6. What do you mean by "aquifer burial conditions"? Does that term refer to the differentiation of "single-layer" and "double-layer" aquifers (but sometimes called "single-aquifer layer" and "double-aquifer layer" instead)?

7. Add an explanation on why atmospheric deposition (4-5 %) was kept in the model even though its contribution is minor, this justifies its inclusion for completeness.

8. The discussion section remains descriptive and does not yet place the new findings in the wider context of existing literature. Please systematically compare your findings with previous studies.

9. I suggest putting forward more specific policy recommendations and summarizing the limitations of the research and its future directions in the discussion.